

# Plasticity-led and mutation-led evolutions are different modes of the same developmental gene regulatory network

Eden T.H. Ng and Akira R. Kinjo

Department of Mathematics, Universiti Brunei Darussalam, Gadong, Brunei

## ABSTRACT

The standard theory of evolution proposes that mutations cause heritable variations, which are naturally selected, leading to evolution. However, this mutation-led evolution (MLE) is being questioned by an alternative theory called plasticity-led evolution (PLE). PLE suggests that an environmental change induces adaptive phenotypes, which are later genetically accommodated. According to PLE, developmental systems should be able to respond to environmental changes adaptively. However, developmental systems are known to be robust against environmental and mutational perturbations. Thus, we expect a transition from a robust state to a plastic one. To test this hypothesis, we constructed a gene regulatory network (GRN) model that integrates developmental processes, hierarchical regulation, and environmental cues. We then simulated its evolution over different magnitudes of environmental changes. Our findings indicate that this GRN model exhibits PLE under large environmental changes and MLE under small environmental changes. Furthermore, we observed that the GRN model is susceptible to environmental or genetic fluctuations under large environmental changes but is robust under small environmental changes. This indicates a breakdown of robustness due to large environmental changes. Before the breakdown of robustness, the distribution of phenotypes is biased and aligned to the environmental changes, which would facilitate rapid adaptation should a large environmental change occur. These observations suggest that the evolutionary transition from mutation-led to plasticity-led evolution is due to a developmental transition from robust to susceptible regimes over increasing magnitudes of environmental change. Thus, the GRN model can reconcile these conflicting theories of evolution.

# INTRODUCTION

The ability to respond to environmental cues by changing the phenotype without changing the genotype is called phenotypic plasticity (*West-Eberhard, 2003*; *Pfennig, 2021*; *Gilbert & Epel, 2009*). Plasticity-led evolution (PLE) is a process in which novel adaptive phenotypes initially induced by a new environment are genetically accommodated over generations (*West-Eberhard, 2003*; *Levis & Pfennig, 2016*; *Levis & Pfennig, 2021*). PLE is characterized by environmental induction of adaptive plastic responses and uncovering of cryptic genetic variation, which are drivers of rapid adaptation (*Ehrenreich & Pfennig,*

Corresponding author
Akira R. Kinjo,
akirakinjo93@gmail.com

*2016*). Therefore, PLE has been proposed to remedy the problem of gradualism implied by the Modern Evolutionary Synthesis (*Laland et al., 2014*; *Ehrenreich & Pfennig, 2016*; *Nishikawa & Kinjo, 2014*; *Nishikawa & Kinjo, 2018*; *Pfennig, 2021*).

In contrast, mutation-led evolution (MLE) refers to evolution initiated by a mutation that induces a novel adaptive phenotype (*Gilbert & Epel, 2009*). Since mutations are random, an adaptive mutation is expected to appear only after a long time, assuming the population has not yet gone extinct. While the Modern Evolutionary Synthesis is necessarily a form of MLE, MLE itself encompasses more general concepts, particularly developmental processes (*Gilbert & Epel, 2009*; *Parsons et al., 2020*) and hierarchical regulation (*Peter & Davidson, 2011*). This article refers to hierarchically regulated developmental processes as hierarchical developmental processes. Our previous work shows that, even without environmental cues, the developmental process is still essential for adaptive evolution in a plausible time scale (*Ng & Kinjo, 2023*). In comparison, it has been observed that quantitative genetics models, not incorporating developmental processes, evolve far more slowly even with environmental cues (*Lande, 2009*; *Scheiner & Levis, 2021*). In this work, we consider a model for MLE that incorporates hierarchical developmental processes.

In our previous work (*Ng & Kinjo, 2023*), we proposed a gene regulatory network (GRN) model that exhibits behaviors compatible with PLE upon large environmental changes (*Levis & Pfennig, 2016*). However, this model is highly robust against mutations and environmental noises in already adapted environments (*Wagner, 1996*; *Espinosa-Soto, Martin & Wagner, 2011b*; *Watson et al., 2014*; *Ng & Kinjo, 2023*). How can the same system be robust in one environment yet exhibit a plastic response to another? This observation suggests there should be a transition from a robust to a plastic regime, depending on the magnitude of environmental change. In the following, we show that this transition manifests as a breakdown of robustness. Furthermore, we demonstrate that the system exhibits MLE under the robust regime and PLE under the plastic regime, thereby unifying two conflicting evolutionary schemes as two modes of the same developmental system.

## MATERIALS AND METHODS

Here, we briefly describe our evolutionary models based on gene regulatory networks. For more details, see our previous article (*Ng & Kinjo, 2023*).

### Environment

In plasticity-led evolution, the environment plays two roles: selector and inducer of phenotypes (*West-Eberhard, 2003*). We assume that these two roles of the environment are highly correlated (*Ng & Kinjo, 2022*; *Ng & Kinjo, 2023*).

We model the environment-as-selector as a 200-dimensional vector **e** of $\pm 1$. This vector is also the optimal phenotype used to calculate the fitness of each individual's phenotype (also modeled as a 200-dimensional vector; see the following subsection). We model the environment-as-inducer as each individual's environmental cue *e*. This vector is obtained by randomly flipping 5% of the elements of **e**.

We obtain a novel environment from a given ancestral environment by randomly flipping a given percentage of the elements of the ancestral environmental vector **e**. Note

that a 50% change in the environmental vector is the maximum change that can be meaningfully achieved in our models. Each element takes $+1$ or $-1$ with equal probability. If we flip 50% of the elements of the environmental vector, then the correlation coefficient between the vectors before and after the change is 0. If we make a change greater than 50%, say 90%, then the correlation coefficient is $-0.8$, which is more significant than the case of 50% change. This implies that the GRN can flip all its state vectors (see below) and find a 90% match to the novel environment without introducing new mutations. In other words, GRNs are more adapted to a novel environment that is 90% different from the ancestral environment than one that is 50% different.

## Development

In the following, we compare two variants (*Ng & Kinjo, 2023*) of the gene regulatory network (GRN) model introduced by *Wagner (1996)*. In these models, the phenotype is a vector, and the genome is a set of interaction matrices. The first model, the Full model, incorporating environmental cues, developmental processes, and hierarchical regulation, exhibits PLE under large environmental changes (*Ng & Kinjo, 2023*). The environmental cue serves as the environmental input to the network. The developmental process is a recursive process that updates the system's internal state, integrating environmental and genetic information into the phenotype. The hierarchical regulation reflects the multilayered nature of biological regulation.

In our developmental GRN models, let $p(s)$ be the phenotype expressed at the $s$-th stage of development and $\tilde{p}(s)$ (and $v(s)$) be the exponential moving average (and variance) of the phenotype. To reflect the hierarchical regulation of GRN elements (such as epigenetic marks, RNAs, and proteins), we introduce layers of 200-dimensional vectors $f(s), g(s)$, and $h(s)$ to represent epigenetic marks, gene expression and higher-order complexes (such as proteins, supramolecular complexes, *etc.*), respectively. For the Full model, we assume the following mutually recursive equations:

$$f_i(s) = \sigma_f \left( \sum_{j=1}^{200} G_{ij} g_j(s-1) + \sum_{j=1}^{200} E_{ij}(e_j - \tilde{p}_j(s-1)) \right),$$

$$g_i(s) = \sigma_g \left( \sum_{j=1}^{200} F_{ij} f_j(s) \right),$$

$$h_i(s) = \sigma_h \left( \sum_{j=1}^{200} H_{ij} g_j(s) + \sum_{j=1}^{200} J_{ij} h_j(s-1) \right), \tag{1}$$

$$p_i(s) = \sigma_p \left( \sum_{j=1}^{200} P_{ij} h_j(s) \right),$$

$$\tilde{p}_i(s) = \alpha p_i(s) + (1-\alpha)\tilde{p}_i(s-1),$$

$$v_i(s) = (1-\alpha)\{v(s-1) + \alpha[\tilde{p}_i(s-1) - p_i(s)]^2\}$$

where $E$ is a matrix that represents environmental regulation of epigenetic marks, $F$ is a matrix representing epigenetic regulation of gene expression levels, $G$ is a matrix

representing genetic regulation of epigenetic marks, $H$ is a matrix representing genetic regulation of higher-order complexes, $J$ is a matrix representing interactions among higher-order complexes, $P$ is a matrix representing regulation of the phenotype. The matrix ensemble $E, F, G, H, J, P$ represents the individual's genome. All matrices are sparse with a density of 0.02. The activation functions $\sigma_f, \sigma_g, \sigma_h$ are based on the inverse tangent functions, and $\sigma_p$ is based on the hyperbolic tangent function. These functions are modified in the spirit of *LeCun (1989)*. The initial conditions are $f(0) = \mathbf{0}$, $g(0) = \mathbf{1}$, $h(0) = \mathbf{0}$ and $p(0) = \mathbf{0}$ where $\mathbf{0}$ is the zero vector and $\mathbf{1}$ is a vector where all elements are equal to 1. We recursively compute the state vectors $f(s), g(s), h(s), p(s)$ for $s = 1, 2, \ldots$ until the total phenotypic exponential moving variance $\sum_i v_i(s) < 10^{-5}$ for some $s \leq 200$ (*i.e.*, the phenotype has converged) and take $\tilde{p}_i(s)$ as the adult phenotype. Otherwise, we say that the phenotype does not converge.

The other model, the *NoCue* model, is identical to the Full model, except it uses the following equation for updating $f_i(s)$ instead:

$$f_i(s) = \sigma_f \left( \sum_{j=1}^{200} G_{ij} g_j(s-1) - \sum_{j=1}^{200} E_{ij} \tilde{p}_j(s-1) \right). \tag{2}$$

The NoCue model does not take in environmental cues ($e_j$). As a result, it does not exhibit phenotypic plasticity and, therefore, serves as a model for MLE.

These models are mathematically equivalent to artificial recurrent neural networks, so they have generic learning capabilities. In this context, adaptive evolution can be viewed as a process of learning the environment by changing the genome (network weights) (*Watson & Szathmáry, 2016*). Notably, the Full model can adaptively respond to environmental cues through development.

**Modeling evolution**

We followed the same procedure as we did in *Ng & Kinjo (2023)* for simulating evolution by applying a genetic algorithm (*Mitchell, 1998*) to a population of 1000 individuals. We impose the role of the environment-as-selector by evaluating the fitness of each individual by matching the macro-environment $\mathbf{e}$ with the adult phenotype $\tilde{p}$ (assuming that $\tilde{p}$ converges). In other words, the macro-environment $\mathbf{e}$ is the optimal phenotype. We also included the number of developmental steps up to convergence into the fitness calculation so that individuals with fewer developmental steps are favored. We define the raw fitness of the $i$th individual as:

$$\omega_i = \exp(-(\alpha \|\tilde{p} - \mathbf{e}\|_1 + \beta N_{\text{step}})) \tag{3}$$

where $\|\tilde{p} - \mathbf{e}\|_1$ is the absolute (L1) distance between the first 40 (out of 200) elements of the adult phenotype $\tilde{p}$ and the corresponding elements of the macro-environment $\mathbf{e}$, $N_{\text{step}}$ is the number of developmental steps until convergence, $\alpha = 20$ and $\beta = \frac{1}{20}$. Only 40 traits (elements of the phenotype vector) are subject to selection, and other $160(= 200 - 40)$ traits are allowed to evolve freely. In this article, we call the value of $\|\tilde{p} - \mathbf{e}\|_1$ *mismatch*. Individuals whose phenotype $\tilde{p}(s)$ does not converge before a pre-specified number (200) of steps are given a zero fitness. The population in which all individuals have completed

the developmental process is called the adult population in the following. The normalized fitness of the $i$th individual of the adult population is given by

$$\Omega_i = \frac{\omega_i}{\max_j\{\omega_j\}} \tag{4}$$

where $\max_j\{\omega_j\}$ is the maximum raw fitness of the adult population. As such, $\Omega_i$ represents a probability of being selected for reproduction.

To generate offsprings, individuals are randomly sampled from the adult population with probability $\Omega_i$. Each sampled individual is returned to the population before another individual is sampled. Selected individuals are paired to become parents. For each pair of parents, two offsprings are produced by randomly shuffling the corresponding rows of the genome matrices with a probability of 50% between the parents. The offspring population is duplicated to make two statistically identical populations of 1000 individuals each. In both populations, each element of the genome matrices of an offspring is independently and randomly mutated with a probability of 0.5% in a way that preserves the density of the genome matrices (*Ng & Kinjo, 2023*). We then let the two populations develop: one offspring population in the "ancestral" environment and the other in the "novel" environment. Only the individuals in the "novel" environment are subject to selection. The population in the "ancestral" environment is used only for comparison and is discarded after measurement. Adaptive evolution of a population is therefore modeled by repeated cycles of development, selection, and reproduction under the "novel" environment.

We let the population evolve in a constant environment **e** for 200 generations. This duration is called an *epoch*. Before simulating for another epoch, we change the environment **e** by flipping a specified proportion of the elements of this vector. We iterate this procedure for many epochs. Before the production runs presented in the Results section, we ran 40 epochs of preparatory runs under 40 completely uncorrelated environments to equilibrate the randomly initialized population and to let the system learn how to respond to environmental changes (*Ng & Kinjo, 2023*). In the following, the novel and ancestral environments refer to the current and last adapted environments, respectively.

## Visualizing evolutionary trajectories

To visualize the evolutionary trajectory of a population, we project the phenotype and genotype of individuals in the population at each generation onto a two-dimensional genotype-phenotype space (*Ng & Kinjo, 2023*). First, the phenotype axis is defined as $\frac{\mathbf{e}_n - \mathbf{e}_a}{\|\mathbf{e}_n - \mathbf{e}_a\|_2^2}$ where $\mathbf{e}_n$ is the first 40 elements of the novel macro-environment, $\mathbf{e}_a$ is the first 40 elements of the ancestral macro-environment and $\|\mathbf{e}_n - \mathbf{e}_a\|_2$ is the Euclidean (L2) distance between $\mathbf{e}_n$ and $\mathbf{e}_a$. We consider only the first 40 out of 200 elements because they correspond to the traits subject to selection. We project the phenotype $p$ of each individual as:

$$\mathbf{p} = (\tilde{p} - \mathbf{e}_a) \cdot \frac{\mathbf{e}_n - \mathbf{e}_a}{\|\mathbf{e}_n - \mathbf{e}_a\|_2^2}. \tag{5}$$

This way, the projected phenotypic values **p** of 0 and 1 correspond to phenotypes perfectly adapted to the ancestral and novel environments, respectively.

Next, the genotype axis is defined as follows. Let $\mathbf{G}_{ij}$ be the vector made by flattening and concatenating all the genome matrices of the $i$th individual of the $j$th generation. Let $\overline{\mathbf{G}}_j = \frac{1}{N}\sum_{i=1}^{N}\mathbf{G}_{ij}$ be the population average genotype vector of the $j$th generation. The genotype axis is defined as $\frac{\overline{\mathbf{G}}_{200}-\overline{\mathbf{G}}_1}{\left\|\overline{\mathbf{G}}_{200}-\overline{\mathbf{G}}_1\right\|_2^2}$. We project the genotype of the $i$th individual of the $j$th generation onto the genotype axis as:

$$g_{ij} = (\mathbf{G}_{ij} - \overline{\mathbf{G}}_1) \cdot \frac{\overline{\mathbf{G}}_{200}-\overline{\mathbf{G}}_1}{\left\|\overline{\mathbf{G}}_{200}-\overline{\mathbf{G}}_1\right\|_2^2}. \tag{6}$$

This way, projected genotypic values $g_{ij}$ of 0 and 1 correspond to the average genotypes before and after one epoch of evolution in a novel environment, respectively.

We call the plot of the projected phenotypic values against the projected genotypic values the genotype-phenotype plot (*Ng & Kinjo, 2023*). Adaptive evolution generally proceeds from the lower left to the upper right on the genotype-phenotype plot.

## Singular value decomposition of cross-covariance matrices

We use cross-covariance matrices to study the correlation between selected phenotypes and environmental noise or genetic variation. We define the Pheno-Cue or Pheno-Geno cross-covariance matrices as

$$C_{ij} = \frac{1}{N}\sum_{k=1}^{N}(p_{ik} - \overline{p}_i)(x_{jk} - \overline{x}_j) \tag{7}$$

where $p_{ik}$ is the $i$th trait of the phenotype vector of the $k$-th individual, $\overline{p}_i$ is the population average of the $i$th trait, $x_{jk}$ is the $j$th environmental factor (for Pheno-Cue) or the $j$th element of the vectorized genome (for Pheno-Geno) of the $k$-th individual and $\overline{x}_j$ is the corresponding population average value.

We apply singular value decomposition (*Yanai, Takeuchi & Takane, 2011*) to the cross-covariance matrix $C$ to obtain orthonormal components as follows (*Navarra & Simoncini, 2010*):

$$C = \sum_i \sigma_i u_i v_i^\top \tag{8}$$

where the superscript $\top$ indicates transpose. In Eq. (8), $u_i$ and $v_i$ are the $i$th left and right singular vectors, interpreted as the $i$th principal axis of phenotypic variation (the left singular vectors) in response to the corresponding principal axis (the right singular vectors) of environmental noises (or genetic variation); $\sigma_i$ is the $i$th singular value, arranged in decreasing order, which represents the cross-covariance between the left and right principal components.

To quantify developmental bias, we used the proportion of the (squared) first singular value to the total cross-covariance:

$$\frac{\sigma_1^2}{\sum_i \sigma_i^2}. \tag{9}$$

The alignment between the first principal axis of phenotypic variation and environmental change is measured by the magnitude of the normalized dot product:

$$\frac{|u_1 \cdot (\mathbf{e}_n - \mathbf{e}_a)|}{\|\mathbf{e}_n - \mathbf{e}_a\|} \qquad (10)$$

where $\mathbf{e}_n$ is the novel environment and $\mathbf{e}_a$ is the ancestral environment.

## RESULTS

In the following, we study evolutionary simulations over various magnitudes of environmental change. We show that the Full model exhibits characteristics consistent with plasticity- or mutation-led evolution when the environmental change is large or small, respectively. We then demonstrate that a breakdown in robustness due to large environmental changes induces adaptive plastic responses and uncovers cryptic genetic variation in the Full model. We show that developmental bias, the skew in phenotype distribution induced by environmental changes, precedes the onset of PLE and is already aligned with environmental changes.

### The Full model exhibits plasticity-led evolution under large environmental changes

This subsection briefly summarizes our previous results (*Ng & Kinjo, 2023*) to introduce fundamental notions describing PLE and MLE. We simulated evolution under large environmental changes for ten epochs, flipping 50% of the elements of the environmental vector at the end of each epoch. We visualized evolutionary trajectories using the genotype-phenotype plots (Figs. 1A, 1B; see also Video S1) (*Ng & Kinjo, 2023*). Phenotypic values of 0 and 1 correspond to phenotypes perfectly adapted to the ancestral and novel environments, respectively. Similarly, genotypic values of 0 and 1 correspond to the average genotypes adapted to the ancestral and novel environments. Generally, the evolution proceeds from the lower left corner to the upper right corner of the genotype-phenotype plot.

For the full model (Fig. 1A), phenotypic values are greater in the novel environment than in the ancestral environment on average, indicating an adaptive plastic response. This response is also called phenotypic accommodation (*West-Eberhard, 2003*; *West-Eberhard, 2005*). The standard deviation in the phenotypic values in the ancestral environment is minimal, indicating the robustness of adapted phenotypes. The standard deviation in the phenotypic values in the novel environment is much greater than that in the ancestral environment in the first generation, indicating an increase in phenotypic variability. Our previous work shows that this is attributed to the uncovering of cryptic genetic variation (*Ng & Kinjo, 2023*). The phenotypic and genotypic values in the novel environment increase somewhat quickly, indicating rapid genetic accommodation. These features of the genotype-phenotype plot assert that the Full model exhibits plasticity-led evolution under large environmental changes.

The NoCue model (Fig. 1B) exhibits neither adaptive plastic response nor uncovering of cryptic genetic variation (by design). The cluttered distribution of points in the lower left corner demonstrates that the early adaptation stage of these trajectories is prolonged.

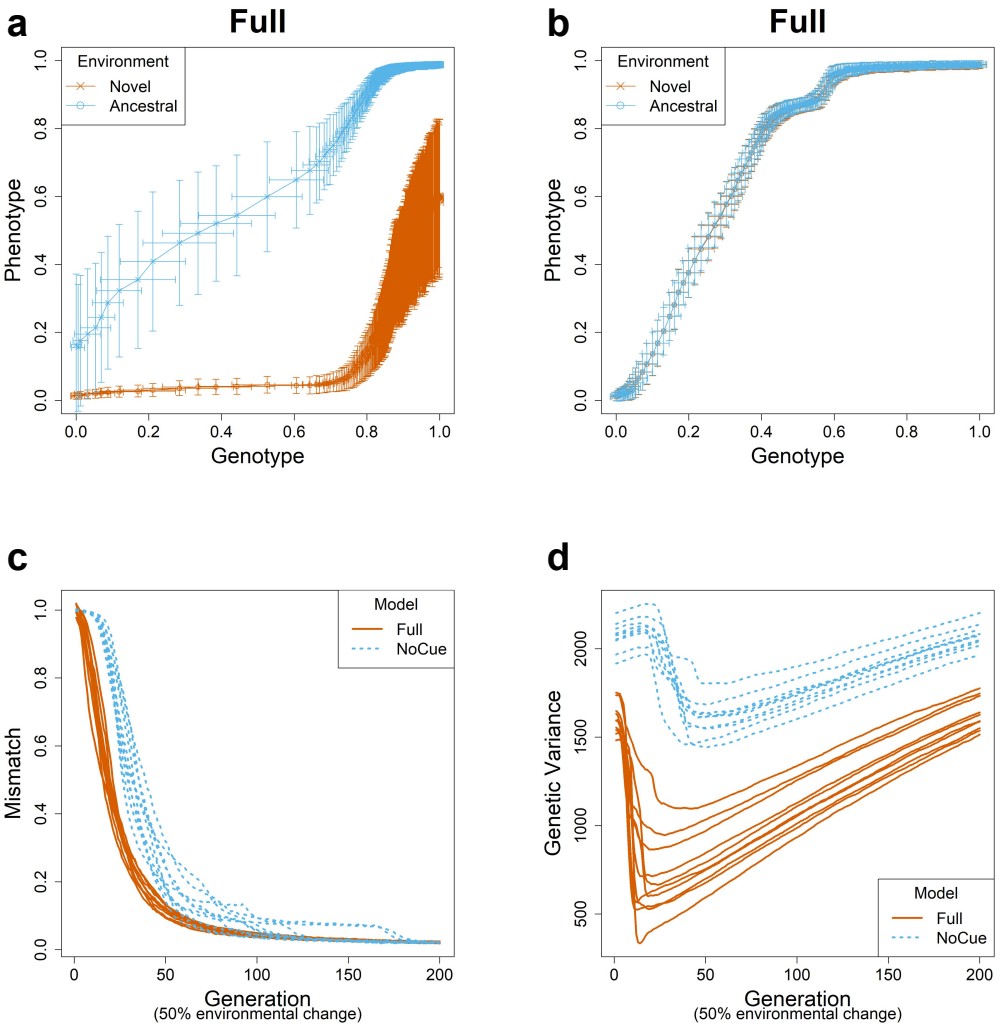

**Figure 1** **Evolution under large environmental changes.** A large environmental change is modeled by flipping 50% of the ancestral environmental vector **e**. The genotype-phenotype plot of an arbitrary epoch of the (A) Full and (B) NoCue model (See also Video S1). Each point (with error bars) represents the population average of genotypic value (horizontal axis) and phenotypic value (vertical axis) after development but before selection at each generation. Cyan and purple represent populations in novel and ancestral environments, respectively. Projected phenotypic values of 0 and 1 correspond to phenotypes perfectly adapted to the ancestral and novel environments, respectively. Projected genotypic values of 0 and 1 correspond to the average genotypes before and after one epoch of evolution in the novel environment. Trajectories of (C) mismatch and (D) genetic variance for ten consecutive epochs. These results were reproduced from our previous work (*Ng & Kinjo, 2023*).

The sparse distribution of points around the center of the plot indicates a rapid change in genotype once adaptive mutations are found.

We next tracked the trajectories of mismatch (Fig. 1C). Recall that the mismatch measures the adaptedness using the distance between the phenotype and environment (see the subsection 'Evolution' in 'Materials and Methods'). The mismatch of the Full model decreases immediately and rapidly. In contrast, the NoCue model exhibits a delay

of around 20–30 generations before the mismatch decreases rapidly, a manifestation of gradualism.

Finally, we checked the trajectories of genetic variance (Fig. 1D). Here, the genetic variance is computed as the sum of the variance in each element of the genome matrices. For the Full model, the genetic variance decreases immediately and rapidly to a minimum before gradually increasing again. The initial rapid decrease corresponds to a purifying selection of uncovered cryptic variations, whereas the later increase indicates the accumulation of cryptic genetic variation. This behavior contrasts the NoCue model, for which the genetic variance gradually increases at the start of each epoch before decreasing to a minimum, after which the genetic variance gradually increases again towards the end of the epoch. The initial increase in the genetic variance coincides with the initial delay of mismatch (Fig. 1C), which is attributed to the time taken to find adaptive mutations in the novel environment. We identify these initial behaviors of the NoCue model as a key signature of MLE.

The genetic variance of the Full model is consistently less than that of the NoCue model (Fig. 1D), indicating that the Full model harbors less standing genetic variation than the NoCue model. Nevertheless, the Full model adapts much more quickly than the NoCue model. This is because most genetic variation remains latent in the NoCue model due to the developmental system's robustness and the absence of environmental cue input.

## The Full model exhibits mutation-led evolution under small environmental changes

We simulated evolution under small environmental changes for ten epochs, flipping 5% of the elements of the environmental vector at the end of each epoch. The evolutionary trajectories of the Full and NoCue models (Figs. 2A, 2B) are nearly identical under small environmental changes. That is, the Full model (Fig. 2A) exhibits neither adaptive plastic response nor uncovering of cryptic genetic variation. The NoCue model exhibits similar evolutionary trajectories under small (Fig. 2B) and large (Fig. 1B) environmental changes. Therefore, under small environmental changes, both the Full and NoCue models exhibit the characteristics of MLE. This observation is consolidated by the trajectories of mismatch and genetic variance (Figs. 2C, 2D).

## The Full model undergoes a sharp transition from MLE to PLE

We have seen that the Full model exhibits PLE or MLE under large or small environmental changes. This observation suggests a transition from MLE to PLE as environmental change increases. To see how this transition occurs, we examined the mean projected phenotype (Fig. 3A) in the first generation after environmental changes of different magnitudes. The mean projected phenotypic value suddenly increases (indicating adaptive plastic response) at around 30% environmental change. The variance in projected phenotype (Fig. 3B) also suddenly increases (indicating the uncovering of cryptic genetic variation) around 30% environmental change.

To further analyze the phenotypic variation under various environmental changes, we examined the cross-covariance matrix between phenotype and environmental cue

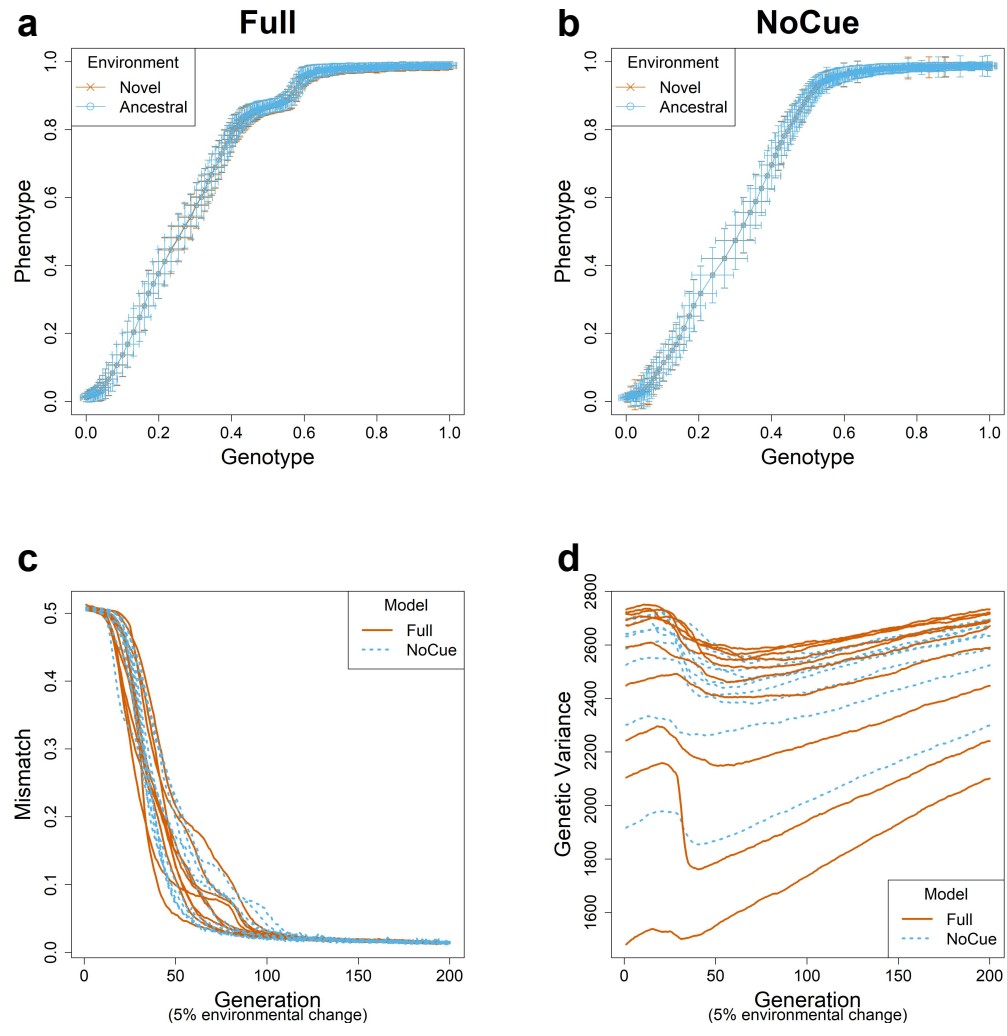

**Figure 2  Evolution under small environmental change.** A small environmental change is modeled by flipping 5% of the ancestral environmental vector **e**. Genotype-Phenotype plots in an arbitrary epoch of the (A) Full and (B) NoCue model (See also Video S1). Trajectories of (C) mismatch and (D) genetic variance for ten different consecutive epochs.

(Pheno-Cue cross-covariance) or genetic variation (Pheno-Geno cross-covariance). These cross-covariance matrices quantify how phenotypic variation is affected by environmental noise or genetic variation. In particular, we only consider the first singular value of the Pheno-Cue (Fig. 3C) and Pheno-Geno (Fig. 3D) cross-covariance matrices as it explains most phenotypic variation. We found that the first singular values also suddenly increase around 30% environmental change. This observation indicates that the phenotypes become susceptible to environmental noise or genetic variation, suggesting a breakdown of environmental and mutational robustness around this point.

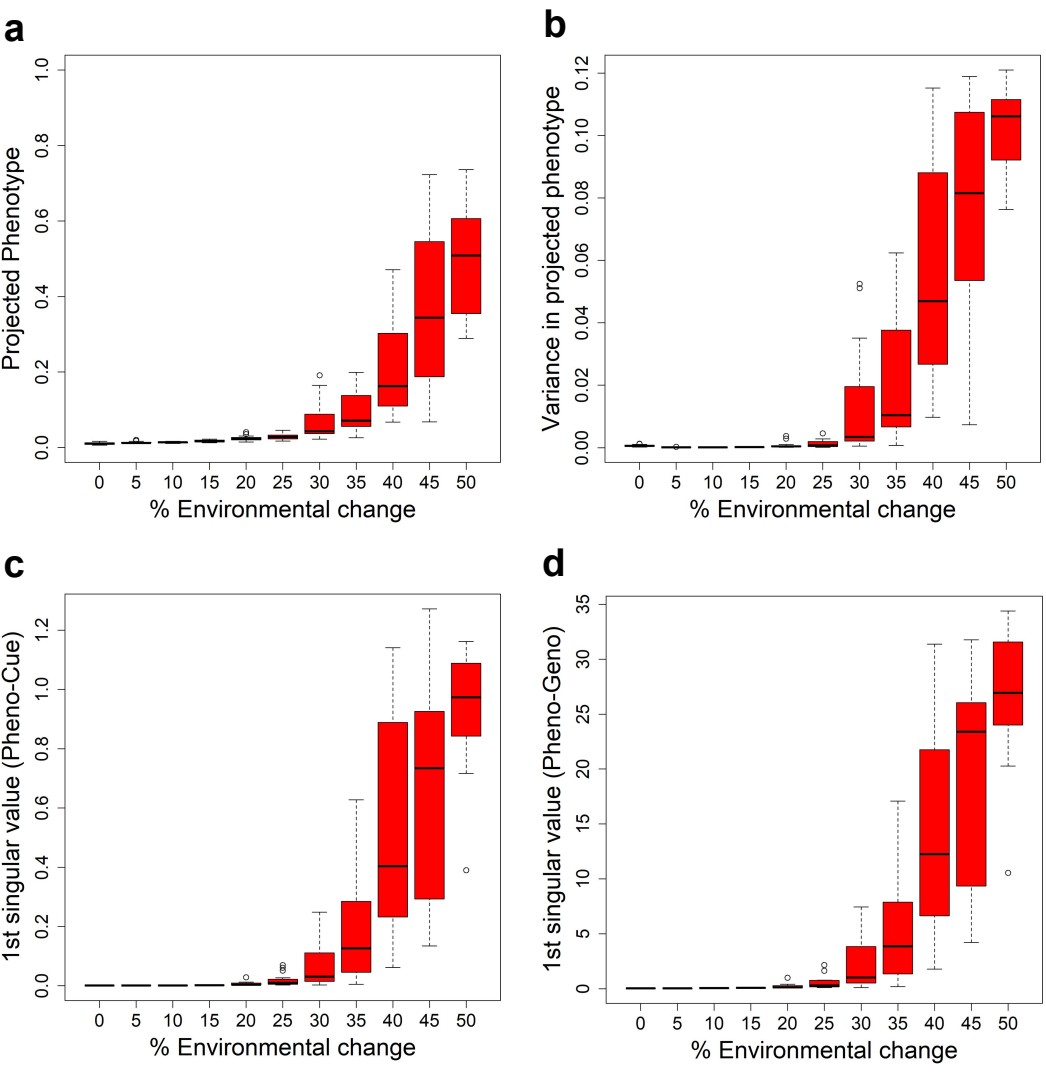

**Figure 3** **Transition from MLE to PLE.** Some characteristic quantities after development and before se-
lection in the first generation when subject to different magnitudes of environmental change for the Full
model. The boxplots were made from the results of 10 epochs. (A) Average projected phenotype. (B) Vari-
ance in projected phenotype. (C) First singular value of phenotype-environmental cue (Pheno-Cue) cross-
covariance matrix. (D) First singular value of phenotype-genome (Pheno-Geno) cross-covariance matrix.

## Developmental bias precedes PLE

The cross-covariance matrices introduced in the previous subsection contain various
information about phenotype distribution, particularly the extent of phenotypic diversity
and the skew in phenotype distribution (*Noble, Radersma & Uller, 2019*). There, we studied
the extent of phenotypic diversity measured by the first singular values. We now study the
skew in phenotype distribution, also called the developmental bias (*Uller et al., 2018*). In the
previous work, we observed large developmental biases aligned with large environmental
changes, whereas such biases were absent in adapted environments (*Ng & Kinjo, 2023*).
To study the transition in developmental bias, we measured the proportion of the first singular value of the Pheno-Cue (Fig. 4A) and Pheno-Geno (Fig. 4B) cross-covariance matrices over environmental changes of different magnitudes. There is an increase in both proportions at around 25% environmental change. Compared to the transition from MLE to PLE (Fig. 3), this increase in bias is less abrupt and occurs at smaller environmental changes. Thus, the developmental bias is more susceptible to environmental changes than the extent of phenotypic diversity. Still, it does not manifest as adaptive plastic responses or increased phenotypic variance due to the system's robustness when the environmental changes are small. When the bias becomes sufficiently large, the system's robustness breaks down, as indicated by an abrupt increase in the cross-covariances (Figs. 3C, 3D).

We next examined the orientation of the developmental bias. Specifically, we measured the alignment between the environmental change and the first phenotype singular vector of the Pheno-Cue (Fig. 4C) or Pheno-Geno (Fig. 4D) cross-covariance matrices over different magnitudes of environmental changes. The alignment is small for less than 25% environmental changes. Above 25% environmental change, developmental bias is well-aligned with environmental change. Thus, the emergent developmental bias facilitates selection by magnifying the difference between the most and least fit individuals. Note that the transition in alignment also happens before the increase in phenotypic variance (see Figs. 3C, 3D). Although plastic responses are invisible at this size of environmental change, such developmental bias aligned with the environmental change is expected to facilitate rapid adaptation should a large environmental change occur (Uller et al., 2018; Noble, Radersma & Uller, 2019).

## DISCUSSION

From an evolutionary perspective, the Full model exhibits plasticity-led evolution (PLE) under large environmental changes and mutation-led evolution (MLE) under small environmental changes. From a developmental perspective, the Full model is susceptible to environmental noise and genetic variation under large environmental changes but is robust under small environmental changes. When the Full model is susceptible to perturbations, adaptive plastic phenotypes are induced, cryptic genetic variation are uncovered, and developmental bias is aligned with environmental change, enabling PLE. When it is robust against environmental and genetic fluctuations, phenotypic variation is suppressed, cryptic genetic variation accumulates, and developmental bias is minimal, necessitating an extensive search for favorable mutations to adapt to small environmental changes, which leads to MLE. Thus, the developmental transition from a robust regime to a susceptible one underlies the evolutionary transition from MLE to PLE.

The developmental transition from robust to susceptible regimes may be considered as the switching of an evolutionary capacitor from the "charging" (accumulating cryptic genetic variation) to "discharging" (purifying uncovered genetic variation) states (Masel, 2013). Indeed, evolutionary capacitance has been suggested to be a generic feature of the GRNs, which are non-linear multi-body feedback systems with gene-environment interactions (Bergman & Siegal, 2003). The present work suggests that switching the

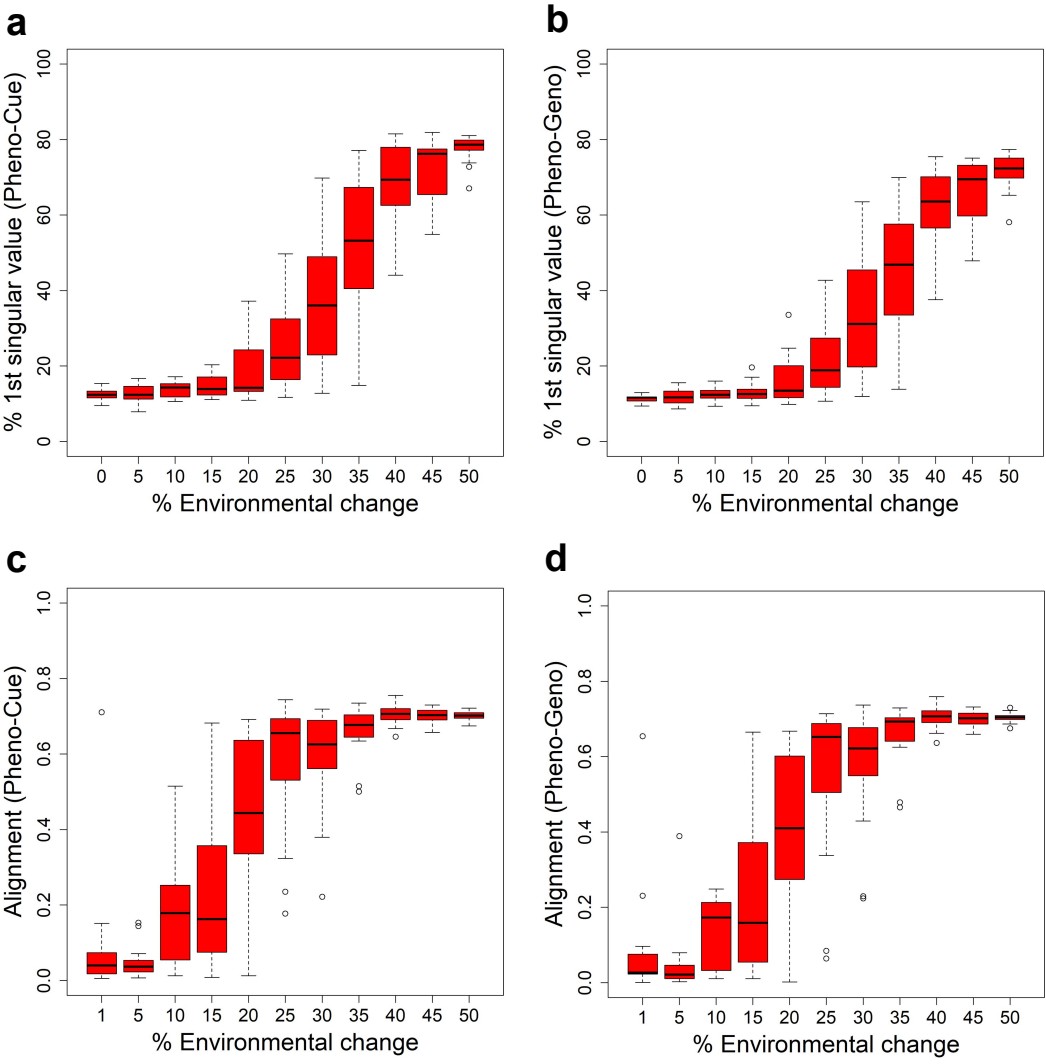

**Figure 4  Developemental bias under various environmental changes.** Measures in the skew and orientation of phenotypic distribution after development and before selection in the first generation when subject to different magnitudes of environmental change for the Full model. The boxplots were made from the results of 10 epochs. Percentage contribution of the first singular value of the Pheno-Cue (A) and Pheno-Geno (B) cross-covariance matrices. Alignment (normalized dot product) between environmental change and the first phenotype singular vector of the Pheno-Cue (C) and Pheno-Geno (D) cross-covariance matrices.

evolutionary capacitor also switches the evolutionary regimes from MLE to PLE. The transition from robust to susceptible regimes over environmental changes implies the existence of a force opposing robustness. According to the Theory of Facilitated Variation (*Gerhart & Kirschner, 2007*), robustness and *adaptability* are (apparently) competing properties of the developmental process. Here, adaptability means the ability of the developmental process to respond to environmental changes adaptively. In our results, adaptability manifests in two ways: (1) adaptive plastic response (Fig. 3) and

(2) developmental bias favorably aligned with environmental change (Fig. 4). In the robust regime, the adaptive plastic response is not visible (Fig. 3), but the developmental bias is already aligned with environmental change (Figs. 4C, 4D). As the magnitude of environmental change increases, the induced adaptability overwhelms the system's robustness, manifests as adaptive plastic responses, and triggers PLE (Fig. 3).

The adaptability of the Full model may appear to be a consequence of its learning capability, as it is mathematically equivalent to an artificial recurrent neural network (*Watson et al., 2014*; *Watson & Szathmáry, 2016*; *Kouvaris et al., 2017*; *Ng & Kinjo, 2022*; *Ng & Kinjo, 2023*). Many studies attempted to explain adaptive plastic response in terms of this learning capability of GRNs (*Watson et al., 2014*; *Watson & Szathmáry, 2016*; *Kouvaris et al., 2017*; *Szilágyi et al., 2020*; *Ng & Kinjo, 2022*). However, these works assume that the novel environment should be sufficiently similar to one of the past environments to exhibit adaptive plastic response and do not consider completely new environments that bear no correlations to any past environments. Thus, the learning theory analogy does not necessarily apply to our results of the Full model. We next argue that the current learning theory analogy is incomplete and requires further extension.

According to *Parsons et al. (2020)*, we should distinguish between two kinds of phenotypic plasticity: active and passive. Active plasticity has been selected through evolution in specific environments and, therefore, anticipates these environments. That means the species already know how to respond adaptively if an environmental cue similar to a past environment is given. Since this response results from selection in the past environment, it is mostly genetically regulated. In other words, active plasticity is a heritable trait. Therefore, it is still in the framework of the Modern Evolutionary Synthesis and does not explain rapid adaptation to entirely new environments. Furthermore, since active plasticity is a trait selected through evolution, it is expected to be a robust process (*Nagata & Kikuchi, 2020*; *Kaneko & Kikuchi, 2022*). This suggests that the variation in the resulting phenotypes is suppressed (*Riley, Oostra & Plaistow, 2023*), hence limiting the efficiency of genetic accommodation. Many simulation studies, as well as most traditional quantitative genetics models, exclusively focus on active plasticity (*Lande, 2009*; *Scheiner, Barfield & Holt, 2020*; *Espinosa-Soto, Hernández & Posadas-García, 2021*; *Watson et al., 2014*; *Nagata & Kikuchi, 2020*; *Kaneko & Kikuchi, 2022*). *Kouvaris et al. (2017)* are motivated by how rapid adaptation in unseen environments is possible, but they still assume significant structural regularities shared across past and present environments.

On the other hand, passive plasticity means an organism responding passively to a new environment that it has never experienced in the species' evolutionary history. As such, the organisms cannot anticipate the new environment, and the role of environmental cues (including noise) becomes relatively more significant. Consequently, the process of the plastic response is not entirely heritable due to the relatively large contribution from the environmental cues (which is definitely not heritable). Furthermore, passive plasticity may involve some randomness, resulting in a larger phenotypic variation. Relatively few simulation studies, including the present work, focus on passive plasticity (*Espinosa-Soto, Martin & Wagner, 2011a*; *Espinosa-Soto, Martin & Wagner, 2011b*; *Nishikawa & Kinjo, 2014*; *Ng & Kinjo, 2023*). Our results show that the randomness appears due to

uncovered cryptic genetic variation. Still, on average, the plastic response is adaptive, albeit imperfect (assuming a strong correlation between the environmental cue and the selective environment). How passive plasticity can be adaptive on average requires further investigation.

In summary, GRNs' adaptability can manifest as active and passive plasticity, but the latter is the key to rapid adaptation to entirely new environments. The current evolution-learning analogy implicitly assumes active plasticity. To fully understand the mechanisms of adaptive plastic response, we need to extend learning theory to account for an equivalent of passive plasticity.

## CONCLUSIONS

We have demonstrated that the gene regulatory network (GRN) model incorporating hierarchical developmental regulation and environmental cues can exhibit plasticity-led evolution (PLE) or mutation-led evolution (MLE) depending on the magnitude of environmental changes. The developmental transition between a robust state and a susceptible state of the developmental GRN drives the evolutionary transition between MLE and PLE. The GRN model presented here may serve as a theoretical basis for understanding evolution, development, and environmental changes in an integrated manner.

## ACKNOWLEDGEMENTS

The authors thank David Marshall, Daphne Teck Ching Lai, and Keiichi Homma for their helpful comments and Haziq Jamil for stimulating discussions.

### Funding
ETHN was supported by the UBD Bursary Award. The funders had no role in study design, data collection and analysis, decision to publish, or preparation of the manuscript.

### Grant Disclosures
The following grant information was disclosed by the authors:
The UBD Bursary Award.

### Competing Interests
The authors declare there are no competing interests.

### Author Contributions
- Eden T.H. Ng performed the experiments, analyzed the data, prepared figures and/or tables, authored or reviewed drafts of the article, and approved the final draft.
- Akira R. Kinjo conceived and designed the experiments, performed the experiments, analyzed the data, authored or reviewed drafts of the article, and approved the final draft.

## Data Availability

The code is available at Zenodo: Akira R Kinjo, & Eden Ng Tian Hwa. (2024). arkinjo/evodevo: Paper2' (Paper2'). Zenodo. https://doi.org/10.5281/zenodo.10607858.

The video is available in the Supplemental File and at Youtube: https://youtu.be/_4x8tDeBhfY.

## Supplemental Information

Supplemental information for this article can be found online at http://dx.doi.org/10.7717/peerj.17102#supplemental-information.

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
