# Peer review of "Plasticity-led and mutation-led evolutions are different modes of the same developmental gene regulatory network"

_PeerJ, doi:10.7717/peerj.17102_

## Round 0.1 · original submission · Minor Revisions

Both reviewers raise several valid issues regarding missing information and the clarity, presentation and interpretation of the results, which should be carefully addressed in a revised manuscript.

Reviewer 1 ·

Basic reporting

The manuscript entitled ‘Plasticity-led and mutation-led evolutions are discrete modes of the same developmental gene regulatory network’ by Ng and Kinjo presents theoretical models that compare the behaviour of GRNs in response to environmental perturbations of different magnitudes. The theoretical models are extensions of on earlier model (Wagner, 1996), and the authors have published on these models previously (Ng and Kinjo, 2023, Scientific Reports). This 2023 publication describes the main insights gained from these models, and the current manuscript further explores one detail of the models, namely how they behave in response to environmental perturbations of different magnitudes.
The authors report their analyses and findings thoroughly and describe relevant aspects in sufficient detail. Large parts of the article are similar to their 2023 publication since the basic settings of the models are identical. While results are generally well reported and concepts are well-explained, I have a few comments that should help the authors to improve the manuscript and clarify some aspects (see under 4. Additional comments).

Experimental design

The theoretical models are almost identical to the ones reported in Ng and Kinjo, 2023, Scientific Reports. In the current article, the authors explore the behaviour of the GRNs in response to environmental changes of varying magnitudes. The models are generally well-explained, but some aspects may require some attention and perhaps some re-analyses.
• For example, it is unclear why the number of developmental steps are included in the calculation of the fitness component. This seems to punish individuals with a more complex development. Why should that be the case?
• In line 115, it is described that the ‘genome matrices of the offspring are randomly mutated’. How exactly was this done, and how much variation did this introduce? I expect that the magnitude of change introduced at this level might have a large impact on the outcomes of the simulations. Has the robustness of the results been tested using a number of different magnitudes of these mutations?

Validity of the findings

In general the reported findings seem valid and the conclusions follow from the results of the simulations. The code for reproducing the results is available on github. Yet, some interpretations seem to be far-reaching; in the Discussion (lines 297-299), it is stated that the ‘Full model exhibits adaptive plastic responses even in entirely new environments’. This is strictly speaking not true since no environments that were more than 50% different to previously experienced ones have been tested. The claim should therefore be modified.

Additional comments

• The authors assume that evolution can occur via two distinct modes: mutation-led and plasticity-led (MLE and PLE). The authors are implicitly assuming that evolution is limited by new mutations appearing. This contrasts to the fact that populations often harbour a substantial amount of standing genetic variation. The authors should at the very least acknowledge this fact, and discuss how this fits with their models and conclusions.
• It is unclear what ‘hierarchical developmental processes’ (line 47) the authors are referring to. This should be define or explain in more detail.
• Some statements do not seem to follow from the results. For example, in lines 179, 180, the authors describe the findings presented in Fig. 1c as a ‘manifestation of gradualism’ for the NoCue model. I would argue that the Full Model shows the same basic pattern and is therefore also compatible with gradualism.
• Regarding the presentation of the findings, I am wondering if the plots could perhaps be improved and presented with more information (the legends are generally extremely short and should contain more information). For example, in Fig. 1a and b, it is not easy to grasp how ‘novel’ and ‘ancestral’ relate to the epochs of different environments described in the Methods section. The legend of the figure should describe what each datapoint exactly is. Also, could the temporal aspect be incorporated, for example by color-coding the datapoints be epoch (and showing novel/ancestral in different plots)? Am I correct in thinking that this would show a gradient from the lower-left to the upper-right corner, or am I misunderstanding the plots? Please provide some more information.
• In relation to this, I am not sure I follow the authors conclusion that the increase in phenotypic variation in the full model in the novel environment is best interpreted as ‘uncovering cryptic genetic variation’ (e.g., lines 143, 165). In my opinion this simply shows that phenotypes are more variable, but there is no need to invoke any notion of genetic variation that has previously been ‘cryptic’ or concealed in some mysterious way. I recommend the authors to remove reference to ‘cryptic mutations’ throughout their manuscript.
• I do not follow the description of the genetic variance changes in Fig. 1d (lines 186-188) with the NoCue model ‘contrasting’ the Full Model: it seems like all curves show the same pattern, albeit the Full Model is generally at a lower level. The statement that genetic variance in the NoCue model is ‘decreasing to a minimum’ is not correct since curves are all increasing towards generation 200.
• Statements such as populations being ‘ready’ (line 250) or ‘preparing for rapid adaptation’ (line 28) seem to imply that evolution proceeds with foresight. I recommend the authors to rewrite these sections.
• I find the Discussion section at times difficult to follow and incoherent. The discussion on genetically encoded plasticity is confusing. In lines 294, 295, it is stated that genetically encoded plasticity is expected to be uniform across a population. First, the authors again do not take standing genetic variation into account (see comment above). Second, I find the term ‘genetically encoded plasticity’ confusing since arguably all plastic traits are influenced by genetic variation in some way, and therefore it is nonsensical to discriminate between genetically encoded and non-encoded plasticity (the latter is ‘developmental plasticity’ in their terminology, which I find very confusing).
• The first and second sentence of the Introduction should be swapped. It is more logical to first explain plasticity, and then plasticity-led evolution.

Reviewer 2 ·

Basic reporting

(0) summary of the paper
The research question of the paper is to clarify under what condition evolution is led by phenotype plasticity (PLE) and under what condition it is led by mutation of the genome (MLE) for change of environmental conditions. The authors introduced a hierarchical gene regulatory network model (GRN) taking into account the effect of environmental conditions on the developmental process. The fitness is defined so that it is high when the expression of the final layer is close to the predetermined environmental value. They conducted the simulation of evolution by genetic algorithm by changing the environmental condition from time to time. They found that MLE with robust phenotype realized when the change of environment change was small, while for the large change of environment, the robustness broke down and PLE realized. Thus the transition between MLE and PLE takes place according to the degree of environmental change.

(1) The paper is written in clear academic English except for a few sentences.

(2) In the Introduction, the background of the research and the research questions are clearly presented,

(3) The paper is organized in a standard manner.

(4) Concerning the figures, the numbers at the axes and the letters in the insets seem to be too small. Descriptions in the figure captions should be a bit more detailed.

(5) I could not find where are the codes and raw data publicized. The authors should open them at some public repository following the journal policy.

Experimental design

(1) The paper presents the original research.

(2) The research question is clearly presented. It is academically meaningful and important.

(3) The hierarchical GRN model taking into account the developmental process is well considered. The computation based on the model seems to be appropriate.

(4) The computational method is well described. But as will be mentioned below, some information is lacking to reproduce the results.

Validity of the findings

(1) The results obtained by the model and the computational method are appropriately presented.

(2) The data needed for concluding are appropriately presented in the figures.

(3) The conclusion is drawn logically from the computational results.

Additional comments

In this paper, an important research question is presented, and the model for exploring it is proposed. The computational process is appropriate, and a reasonable conclusion is drawn based on the computational result. Therefore, I recommend it to be published in PeerJ Journal after revision. My concerns are listed below adding to already mentioned points.

(1)Major point
1. Although the transition from MLE to PLE is rapid, there seems to be no clear threshold, and thus the transition seems to be gradual in a strict sense (Although there is a value of environmental change characteristic to the transition). In this context, while the section title at L.203 is "phase transition," I think there is no clear transition point. If there is no clear threshold between MLE and PLE, the term "phase transition" should be used carefully. My opinion is that, in terms of physics, what we observe is not a phase transition but a "crossover phenomenon." This observation is consistent with the fact that the value of the environmental change that the alignment varies largely and the value of the environmental change that the projected phenotype rapidly increases are different. In this connection, the term "transition point" in L.220 should also be reconsidered. The term "discrete mode" in the title should also be reconsidered.

2. Among the results shown in Fig.1 a and b, I did not understand how the ancestral environment condition was computed. According to the description in the text, the environment changes at the end of the epoch; thus, it seems that only adaptation to the novel environment can be computed. More explanation is required about this point.

3. I did not understand how to calculate the average genotype in the genotype-phenotype plot.

(2)Minor points
1. L.94 Although details of the model are described in the previous paper, the form of the activation function should be given for the self-containedness of the paper.

2. L.108 "adult phenotype p" should be p-tilde?

3. L.112 I did not understand what "with replacement" means.

4. L.115 The mutation rate should be given for the readers to reproduce the results.

---

## Round 0.2 · accepted · Accept

I have assessed the revised manuscript myself. The authors have carefully addressed the reviewers' comments. The manuscript has improved and is now ready for publication.